# Facile 3D Boron Nitride Integrated Electrospun Nanofibrous Membranes for Purging Organic Pollutants

**DOI:** 10.3390/nano9101383

**Published:** 2019-09-27

**Authors:** Dai-Hua Jiang, Pei-Chi Chiu, Chia-Jung Cho, Loganathan Veeramuthu, Shih-Huang Tung, Toshifumi Satoh, Wei-Hung Chiang, Xingke Cai, Chi-Ching Kuo

**Affiliations:** 1Institute of Organic and Polymeric Materials, Research and Development Center of Smart Textile Technology, National Taipei University of Technology, Taipei 10608, Taiwan; d06549013@ntu.edu.tw (D.-H.J.); chie92714@gmail.com (P.-C.C.); ppaul288@yahoo.com.tw (C.-J.C.); anloga947715@gmail.com (L.V.); 2Institute of Polymer Science and Engineering, National Taiwan University, Taipei 106, Taiwan; shtung@ntu.edu.tw; 3Faculty of Engineering, Hokkaido University, Sapporo 060-8628, Japan; satoh@eng.hokudai.ac.jp; 4Department of Chemical Engineering, National Taiwan University of Science and Technology, Taipei 10607, Taiwan; whchiang@mail.ntust.edu.tw; 5Institute for Advanced Study, Shenzhen University, Shenzhen 518060, Guangdong, China; cai.xingke@szu.edu.cn

**Keywords:** boron nitride, electrospun fibers, removal of pollutants, 3D structure

## Abstract

Elegant integration of three-dimensional (3D) boron nitride (BN) into the porous structure of a polymer nanofiber (NF) membrane system results in a surface with enhanced absorption capacity for removal. Various BN-based applications were designed and developed successfully, but BN-based absorption systems remain relatively unexplored. To develop a reusable absorption strategy with high removal efficiency, we used a composite of 3D BN and polyacrylonitrile (PAN) to prepare a NF membrane with a porous structure by using electrospinning and spray techniques (BN-PAN ES NFs). The removal efficiency of the 3D BN NF membrane was higher than that of a pure carbon NF membrane. Water pollutants, such as the dyes Congo red (CR), basic yellow 1 (BY), and rhodamine B (Rh B), were tested, and the absorption ratios were 46%, 53%, and 45%, respectively. Furthermore, the aforementioned dyes and pollutants can be completely eliminated and removed from water by heating because of the high heat resistance of 3D BN. The membrane can be recycled and reused at least 10 times. These results indicate that BN-PAN ES NFs have can be used in water purification and treatment for absorption applications, and that they can be reused after heat treatment.

## 1. Introduction

Growth of industrial activity and increasing water usage (inevitable water utilities) around the world have led to the release of various pollutants, such as toxic heavy metals and organic pollutants into the aquatic environment, particularly in developing countries. Organic pollutants include phenols, dyes, pesticides, detergents, insecticides, herbicides, and other persistent organic pollutants. Organic pollutants are a major concern because of their potential mutagenicity, carcinogenicity, teratogenicity, and high bioaccumulation. In addition to biological and human hazards, organic pollutants permanently damage land and rivers. Therefore, recycling and reusing water are becoming increasingly important for sustainable utilization of water resources and protection of the environment [1,2,3,4,5]. Many treatment methods are currently available for removing organic pollutants from water, such as reverse osmosis [6], adsorption [6,7,8,9], chemical precipitation [10], coagulation [11,12], electroplating [13], oxidation–reduction [14], and ion exchange [15,16]. Various water treatment methods use adsorption because it is economical, universal, and easy to operate [1,2,3,4,5]. Adsorption-based water treatment methods use materials with a high surface/volume ratio or special manufacturing methods to achieve remarkably high removal of organic pollutants.

Electrospinning is an easy, versatile, and inexpensive technique to produce nanometer-scale fibers for assembling various functional nanofibers (NFs) [17,18,19]. We successfully prepared various ES polymer NFs for different applications, such as sensing of pH levels [20], temperature [20,21], NO level [22], and heavy metal ions [23,24]. Cleaning toxic pollutants by using a high surface/volume ratio of ES NFs is an important subject. Some studies have suggested that conducting heterogeneous photocatalysis by using zinc oxide (ZnO) NFs is a new promising technique for cost-effective treatment of organic pollutants and transformation of hazardous substances into their benign forms. In general, heterogeneous photocatalysis by using ZnO NFs involves developing smart approaches to reduce the harmful effects of highly toxic and recalcitrant pollutants. Some research has focused on the fabrication of ES hybrid NFs or composite ES NFs and their applications in the photodegradation of different organic pollutants that are discharged into wastewater in textile and other industries [25,26].

As a synthetic material, boron nitride (BN) has several well-defined crystallographic structures, such as cubic BN (c-BN), hexagonal BN (h-BN), rhombohedral BN (r-BN), and wurtzite BN (w-BN). BN has many special properties, such as low density, low conductivity, electrical insulating properties, excellent mechanical properties, a high surface area, extremely high resistance to oxidation, high chemical stability, a low thermal expansion coefficient, a high melting point, and high thermal conductivity [27,28,29]; hence, it has been widely used as an electrical insulator and in heat resistant materials for decades because it is a good conductor of heat as well as a good electrical insulator. High-quality h-BN crystal prepared at 2100 °C by using a nickel-molybdenum solvent was found to be a promising deep ultraviolet light emitter [30]. r-BN was detected during the synthesis of h-BN. r-BN was reported to be formed during the conversion of c-BN to h-BN [27]. h-BN is a two-dimensional (2D) layered material that is stable at 1500 °C in air and does not react with most chemicals. h-BN has been morphed to manufacture products for various applications, including BN powders, one-dimensional (1D) nanotubes (BNNTs), 1D BN NFs, 2D nanosheets, 2D BN porous belts, three-dimensional (3D) particles, and 3D BN structures. Several studies on 3D BN structures have successfully combined the properties of BN materials and the stability of 3D nanostructures to produce materials with high adsorption capacity and stable cycle. Many potential applications, including as a photocatalyst [31,32], catalytic supports [33,34], filtration [35,36], and organic pollutant adsorption [37,38] have been demonstrated. In a study by Dan Liu’s et al presents direct synthesis of 3D BN structures by using a simple thermal treatment process. A 3D BN structure consists of an interconnected flexible network of nanosheets. The typical nitrogen adsorption/desorption results demonstrated that the maximum specific surface area of the as-prepared samples was 1156 m^2^·g^−1^, and the total pore volume was approximately 1.17 cm^3^·g^−1^. The 3D BN structure displayed high adsorption rates and large capacities for organic dyes in water without any other additives because of its low density, high resistance to oxidation, high chemical inertness, and high surface area. Notably, 88% of the starting adsorption capacity was maintained after 15 cycles. These results indicated that the potential application of 3D BN structures as an environment-friendly material for water purification and treatment [29]. However, thus far, no study has combined the high surface-to-volume ratio of ES NFs and the porous structure of 3D BN for application in water purification and treatment.

In view of aforementioned advantages, we propose that 3D BN is a suitable material to be used in absorption, and it can be combined with ES NFs. First, we produced ES hybrid or composite ES NFs using BN-PAN materials. After absorbing dyes and pollutants from water, the NFs can be reused after subjecting them to heat treatment because 3D BN has high heat resistance. Our study is the first to demonstrate the adsorption application of materials prepared by using ES. In this study, we combined the 3D BN structure and ES NFs to prepare high adsorption material with the novel electrospinning spray coating composite method and demonstrated the dye scavenging applications as a proof of concept.

## 2. Experimental Section

### 2.1. Materials

Polyacrylonitrile (PAN, 98%, Mw = 150,000), basic yellow (BY, C_28_H_31_C_l_N_2_O_3_) and Congo Red (CR, C_32_H_22_N_6_Na_2_O_6_S_2_, ≥35%) were purchased from Sigma-Aldrich, St. Louis, MO, USA. Dimethylformamide (DMF), C_3_H_7_NO (98%), ethanol (95%), and methanol (HPLC grade) were purchased from ECHO Chemical Co., Miaoli, R.O.C. Boron Oxide (B_2_O_3_) and urea (CH_4_N_2_O) were prchased from J.T.Baker-Avantor Chemical Co., HsinChu, R.O.C. Rhodamine-B (Rh B, C_28_H_31_C_l_N_2_O_3_) was purchased from Acros Organics Co., Beijing, China.

### 2.2. Preparation of Electrospinning Nanofibers (NFs) and Polymer Solution

The PAN solution was placed in a 5 mL syringe. The PAN was first dissolved in the solvent, N, N-dimethyl formamide (DMF), to make it into a polymer solution which has a concentration of 12 wt%. The feed rate of the solution was 1 mL·h^−1^. The metallic needle was connected to a high-voltage power supply, and a piece of aluminum foil was placed 20 cm below the tip of the needle to collect the NFs. The spinning voltage was set at 12.9 kV. After collection time of 10 min, Non-woven electrospun PAN NF was collected on aluminum foil. All experiments were carried out at room temperature and at a relative humidity of about twenty percent.

### 2.3. Synthesis of Three-Dimensional Boron Nitride Precursor

In a typical synthesis, boron trioxide (B_2_O_3_) and urea (CH_4_N_2_O) with 1:10, 1:5, and 1:1 molar ratios were respectively mixed in 3 mL methanol under stirring to form a clear colorless solution, which was prepared according to the reported procedures [29]. 

### 2.4. Regulation of Spray Coating

The different molar ratios of BN and urea (1:10, 1:5, and 1:1) were prepared and utilized as a precursor for the spray coating process to form 3D BN. The sample was placed 20 cm below the tip of the spray nozzle to do spray coating. The pressure supply was about 1 kg·cm^−2^, and the flow rate was controlled at about 0.5 mL·h^−1^. The spraying times of all the samples were 10, 30, and 60 s, respectively.

### 2.5. Process of Electrospun Boron Nitride PAN Nanofiber(NF) Membrane

We followed two different mechanisms to manufacture the BN ES NFs in our research which is illustrated in Figure 1. The spray coating collecting time of all sample were fixed as 30 seconds. The collecting time of the electrospun fibers was about 10 min. In this part, we compared two techniques of the spray coating to collect the boron nitride precursor, and they were respectively (1) direct spraying and (2) electrospinning and spraying composite technology. Then, the prepared samples were kept in the heating oven to dry, and we used the SEM to observe the distribution of BN precursor on the NF membrane. One of the methods was direct spray coating. We took the as-spun PAN NF membrane as the substrate and used the spray jet to spray the BN precursor solution on the surface of the membrane. Then, the prepared samples were kept in the heating oven to dry.

### 2.6. Process of Electrospinning 3D Boron Nitride Carbon NF Membrane

Electrospun carbon NFs (ECNFs) were prepared by electrospinning PAN solution followed by stabilization and carbonization. In the stabilization process, the PAN NF membrane was placed in the quartz boat and heated at a rate of 1 °C·min^−1^ from 30 °C to 280 °C and held there for 1 h in a constant flow of air. Carbonization was carried out by heating the stabilized PAN NF membrane in a constant flow of nitrogen/hydrogen (10% hydrogen) gas to 800 °C at a rate of 5 °C·min^−1^ and maintained at 800 °C for 2 h. In the stabilization treatment process of electrospinning PAN NFs, the BN precursor also become porous BN nanosheets with a 3D structure at the same time. During the heat treatment process, the urea is decomposed at high temperature, releasing a number of gaseous species including CO_2_, HCNO, N_2_O, H_2_O, and NH_3_. The escape of gases (gas bubbles) acts as fugitive templates of the final porous structure. The white crystal powders were heated in a tube furnace at 800 °C for 2 h under nitrogen/hydrogen (10% hydrogen) gas flow, and a black sample was produced.

### 2.7. Characterization of 3D Boron Nitride Carbon NF Membrane

The structure of the sample was observed with a Hitachi H-7100 (operating at 100 kV) transmission electron microscope (TEM) (Hitachi H-7100, Tokyo, Japan), and scanning electron microscopic (SEM) (Hitachi TM-3000, Tokyo, Japan) images were obtained with a cold-field emission scanning electron microscope (HR-SEM) (Hitachi S-4800, Tokyo, Japan) equipped with energy-dispersive. The X-ray photoelectron spectra (XPS) (ESCALAB 250, England) were collected on an ESCA Lab MKII X-ray photoelectron spectrometer using non-monochromatized Mg-Ka X-ray as the excitation source.

### 2.8. Dye Removal Test

Dye solutions (CR, BY, and Rh B) of different concentrations were prepared by dissolving appropriate amounts of CR into deionized water (DI water), respectively. In a typical adsorption of a CR experiment, 3 × 3 cm of the 3D BN NF membrane was added to 100 mL CR aqueous solution (10 mg/L), and under stirring, and UV-vis absorption spectra were recorded at different time intervals to monitor the process at 496 nm. The adsorption isotherm was obtained by varying the initial CR concentration. The adsorption studies of Rh, B, and BY were similar to those of CR except for the detection wavelength difference (553 nm for Rh B and 412 nm for BY).

## 3. Results and Discussion

### 3.1. Chemical and Structural Characterization of Morphology

The morphology was modified by selecting suitable solutions during ES. Figure 2 shows the cold field emission scanning electron microscope and energy dispersive spectrometer (FE-SEM) images of PAN ES NFs with different magnifications for 12 wt% concentration. As seen in Figure 2, no beads were observed on the surface or layer of the PAN NFs membrane. Stable PAN NF membrane generation can be achieved by regulating parameters and environmental factors. Figure 2 shows that the diameters of all NFs in the PAN NF membrane are similar and the average diameter of the NFs is about 540 nm.

Spray coating were adjusted with different spraying times of 10, 30, and 60 s, to improve the fiber quality. The collecting time of electrospinning is about 10 min. In this study, we used direct spraying to collect the BN precursor. Then, the collected sample was placed in the heating oven to dry, and we used the FE-SEM to observe distribution of the BN precursor on the NF membrane (Appendix A). The FE-SEM image shows the different spray coating times of the BN ES NF membrane. In the spray coating process, the spraying time was set at 10, 30, and 60 s. An air jet was used to spray the precursor solution, and it was collected on the surface of the NFs. Appendix A shows the low-magnification image when fewer BN sheets were placed on the PAN NFs. We observed that BN was thinner and smaller than the other materials. Appendix A shows that numerous BN sheets were placed on the NF membrane, and evidently, the BN sheets connect and stack on each other and attach the PAN NFs to form a large BN film on the PAN NF membrane. The successive stacking of BN structures with increased spray coating time of 60s was clearly evidenced with Appendix A. Such thick films apparently lose their appealing flexibility and turn fragile under higher spray time. Therefore, we concluded that 30 s was the most suitable spraying time for the BN ES NF membrane in the spray coating process.

In our study, we compared two techniques of spray coating to collect the BN precursor, namely direct spraying as well as electrospinning and spraying composite technology (Figure 1). Figure 3 shows the SEM images of spray coating with different spraying techniques. Figure 3a,b show the SEM images of direct spraying. As shown in Figure 3a, the BN precursor covered the NF surface. Furthermore, we observed the NF membrane layer and noted that the BN precursor could not seep into the middle or bottom of the NF membrane. The second technique used composite technology, electrospinning, and spraying to prepare the BN ES NF membrane. In this method, the coating was directly sprayed on the NFs during electrospinning to produce the PAN NF membrane. Simultaneously, the BN precursor solution was sprayed on the NFs and the BN precursor was collected along with the PAN NFs. By using this method to manufacture the BN ES NF membrane, the BN precursor can be placed in the middle or at the bottom of the PAN ES NF membrane. These results are shown in Figure 3c,d. Notably, the BN ES NF membranes prepared using the electrospinning and spraying composite technique are better because NFs protect and keep BN inside the NF membrane layers.

To obtain a high-quality 3D BN structure, we used varying ratios of the BN precursor in spray coating, as shown in Appendix A. In this study, we used boron trioxide (B_2_O_3_) and urea (CH_4_N_2_O) as precursor materials in the precursor solution. B_2_O_3_ and CH_4_N_2_O were stirred in molar ratios of 1:10, 1:5, and 1:1 in 3 mL methanol to obtain a clear, colorless solution. To compare the effect of different ratios of the BN precursor, we can observe the distribution is very different from low-magnification SEM images, as shown in Appendix A, and they indicated that boron oxide and urea were mixed in the ratios of 1:10, 1:5, and 1:1. In the precursor synthesis process, the areas of the BN precursor film increased with the increase in the boron oxide ratio, which led to a higher ratio of boron oxide to urea to form the BN precursor crystal. 

Figure 4a show the SEM images of the pure PAN NF membrane, and the inset of Figure 4a shows the exterior of the PAN NF membrane. In Figure 4a, we observe that the NF membrane appears white and shiny before it is subjected to the heat treatment process and the surface of the PAN NFs is smooth. The average diameter of the PAN NFs is approximately 540 nm. The PAN NF membrane transforms into carbon NF membrane when it is subjected to heat treatment, as shown in Figure 4b. The inset of the Figure 4b shows the exterior of the carbon NF membrane, which appears black and is not shiny. The carbon NF surface is coarser than that of the PAN NF, and the average diameter of the carbon NFs is approximately 340 nm. The differences between the PAN and carbon NFs are attributable to the long-chain polymer structure dehydrogenation and cyclization which form an orderly row-column cyclic structure. After crossing the carbonization temperature of 625 °C, the PAN NFs ultimately undergo carbonization and a subsequent denitrification process. 

Next, we investigated the effects of different molar ratios on BN after heat treatment, and the distribution is presented in Figure 5a,c,e; they respectively indicated that boron oxide and urea were mixed in the ratio of 1:10, 1:5, and 1:1. It is already revealed that the spray coated precursors can lead to the film formation b/w the NF layers [39]. The areas of the BN precursor film increased with the increase in the amount of boron oxide. The result is presented in Figure 5a,c,e. According to the dynamic templating approach, byproducts of the condensation of the solid framework evolve as nanobubbles that act as fugitive templates of the final porous structure [28]. Different molar ratios of the BN precursor in the synthesized 3D BN structures are shown in Figure 5. We find that samples with BN molar ratios of 1:10 and 1:5 have a typical 3D porous structure. When boron oxide and urea bind, the samples with BN molar ratios of 1:10 and 1:5 had excess urea on the white BN crystal, which formed the fugitive templates. As shown in Figure 5a, there are fewer 3D BN porous structures on the samples with a BN molar ratio of 1:10, which may cause low efficiency in the absorbance test. The sample with the BN precursor molar ratio 1:1 had less leftover urea because most of the urea was bound with boron oxide to form a larger white crystal. Therefore, during the heat treatment process, urea decomposed at high temperature to release gaseous species to form the 3D BN porous structure and there was less dynamic templating. The sample with the BN precursor molar ratio of 1:1 had a large 3D BN structure because of binding in the BN precursor stack. At this point, we decided that the BN precursor molar ratios of 1:5 are the best option to be utilized in the dye test. Corresponding surface elemental analyses of the pure carbon and 3D BN NF membranes were conducted using energy-dispersive X-ray (EDX) spectrum, as shown in Appendix A. Show the EDX of pure PAN carbon NFs and BN-PAN composite ES NFs, respectively. The results indicate that the pure carbon NF membrane consists of elements C and O, as shown in Appendix A. In particular, the signal of C is strong. The EDX spectrum in Appendix A indicates that the 3D BN NF membrane consists of elements C, O, B, and N. In particular, the elemental analyses represent different areas of the BN ES NF membrane, and we can clearly distinguish the areas with BN and carbon structures. We can conclude that we successfully prepared a 3D BN NF membrane by using the electrospinning and spraying composite method.

To improve our understanding of the porous structure of our prepared fibers, additional SEM and TEM images are measured. Figure 6a shows the FE-SEM image of the 3D BN porous structure. When boron oxide and urea were binding, the samples with the BN molar ratio of 1:5 had excess urea on the white BN crystal, which formed fugitive templates. The TEM images in Figure 6b,c clearly show that 2D BN nanosheets stack to form the 3D porous structures. XPS were recorded, and the chemical states of B and N elements were investigated using XPS. Appendix A shows typical B1s, N1s, O1s, and C1s spectra, with corresponding binding energies of 190.8, 398.4, 531, and 284 eV, respectively. These values are very close to the previously reported values of BN layers with BN_3_ and NB_3_ trigonal units [28,40,41]. From the rational consecutive optimization done with a mechanism, spray coat time, and BN precursor ratio, we prepared the 3D BN NF membrane with a good porous architecture.

### 3.2. Dye Test of CR, BY, and RhB

Dye solutions (CR, BY, and RhB) of different concentrations were prepared by dissolving appropriate amounts of CR, BY, and Rh B in deionized water. In a typical adsorption of CR experiment, 3 × 3 cm^2^ of the as-prepared BN sample (496 nm) was added to 100 mL CR aqueous solution (10 mg/L) under stirring, and UV-visible (UV-vis) absorption spectra were recorded at different time intervals to monitor the process. The adsorption isotherm was obtained by varying the initial CR concentration. The adsorption studies of BY and Rh B were similar to those of CR except for the difference in detection wavelengths (412 nm for BY and 553 nm for Rh B). CR, an anionic dye, is regarded as a primary toxic pollutant in water resources. UV-vis absorption spectroscopy was used at the maximum absorption wavelength of CR (496 nm) to estimate the adsorption kinetics in an aqueous CR solution (Figure 7a). BY, a textile dye (cationic), regarded as a primary toxic pollutant in water resources, was chosen as a typical organic waste. The initial BY concentration in water was 50 mg/L, as shown in Figure 7b. The characteristic absorption of BY at 412 nm was chosen to monitor the adsorption process. Rh B dye is commonly used tracer in determining the flow and transport. Rh B dyes fluoresce and can thus be detected easily and inexpensively by using fluorometers. Rh B dye finds extensive roles in biotechnology applications, such as fluorescence microscopy, flow cytometry, and fluorescence correlation spectroscopy. UV-vis absorption spectroscopy was used at the maximum absorption wavelength of Rh B (553 nm) to estimate the adsorption kinetics in a 10 mg/L aqueous Rh B solution, as shown in Figure 7c. In the Figure 7a–c, we can clearly observe the absorption-intensity of dye solutions (CR, BY, and RhB) without NF membrane is sustained, directing dye solutions is not degradable in the air condition for 180 min.

Figure 8 shows the UV–vis absorption spectra and the removal efficiency of the pure carbon NF membrane in aqueous solutions of CR, BY, and Rh B. After 180 min, the carbon NF membrane removed approximately 13% of the CR, 30% of the BY, and 16% of the RhB, as shown in Figure 8. In this study, we employed 3D BN NF membrane aiming to cohesive improvement with adsorption capacity and higher surface/volume ratio. Figure 9a–c show the adsorption spectra of the 3D BN NF membrane, and that the absorption of all dye samples is proportional to the time. Notably, the differences between the pure carbon and 3D BN NF membranes can be investigated by comparing adsorption rates, the absorption ratios of CR, BY, and RhB are 46%, 53%, and 45%, respectively. By contrast, the 3D BN NF membrane demonstrated efficient absorption compared with the pure carbon NF membrane because of the improved surface to volume ratio aided with formation of porous structured nanosheets. Comparing Figure 8c to Figure 9c, it was observed that the BN modified sample performed much better for RhB aqueous solutions. This means that the porous architecture of 3D BN for BN modified sample enhanced the absorption ability, comparing with pure carbon NF membrane. Compared with Figure 9b,c, a dramatically drop at 150 min is observed in Figure 9c. This may have been caused by different adsorption abilities between 3D BN NFs and different dye solutions. The slope of the absorption vs. time have a little bit different, but the final result of the absorption rate is almost the same. Appendix A shows that the absorption ratios of Rh B researched at 90%, indicating almost full complete removal of dye over time (450 min). Furthermore, in our study, we immersed our sample to the dye solution to absorb the dye and then removed the dye by heating to 500–800 °C. Most dyes were carbonized and degraded by heating to such high temperature. So, we removed it out, and the membrane could be reused again. As shown in Appendix A, the membrane can be recycled and reused at least 10 times. Our interesting results suggests that simple optimized blending ratio of BN precursors on PAN nanofibers can produce desirable high adsorption capacity using the combined electrospinning and spray coating technique.

## 4. Conclusions

In this study, we successfully fabricated a PAN NF membrane with a 3D BN porous structure by using a simple electrospinning and spraying composite technique. After heat treatment, the 3D BN with a porous structure located within PAN NFs was successfully framed. By using sensible optimization, we determined that the optimal parameters of the precursor are a molar ratio of 1:5 and a spraying time of 30 s. Furthermore, the prepared BN-PAN ES NF membrane showed a higher adsorption ability of the tested dyes from water ascribed to the 3D BN nanosheets successful embedment onto NFs with good porous architecture. The developed 3D BN material has better porous structure, excellent adsorption towards anionic, cationic, and fluorescent dyes, and is lightweight. These credible features make the 3D BN ES NF membranes suitable for water purification, and they have a greater wide range of potential applications, such as in filters, energy storage, hydrogen storage, and capacitors.

## Figures and Tables

**Figure 1 nanomaterials-09-01383-f001:**
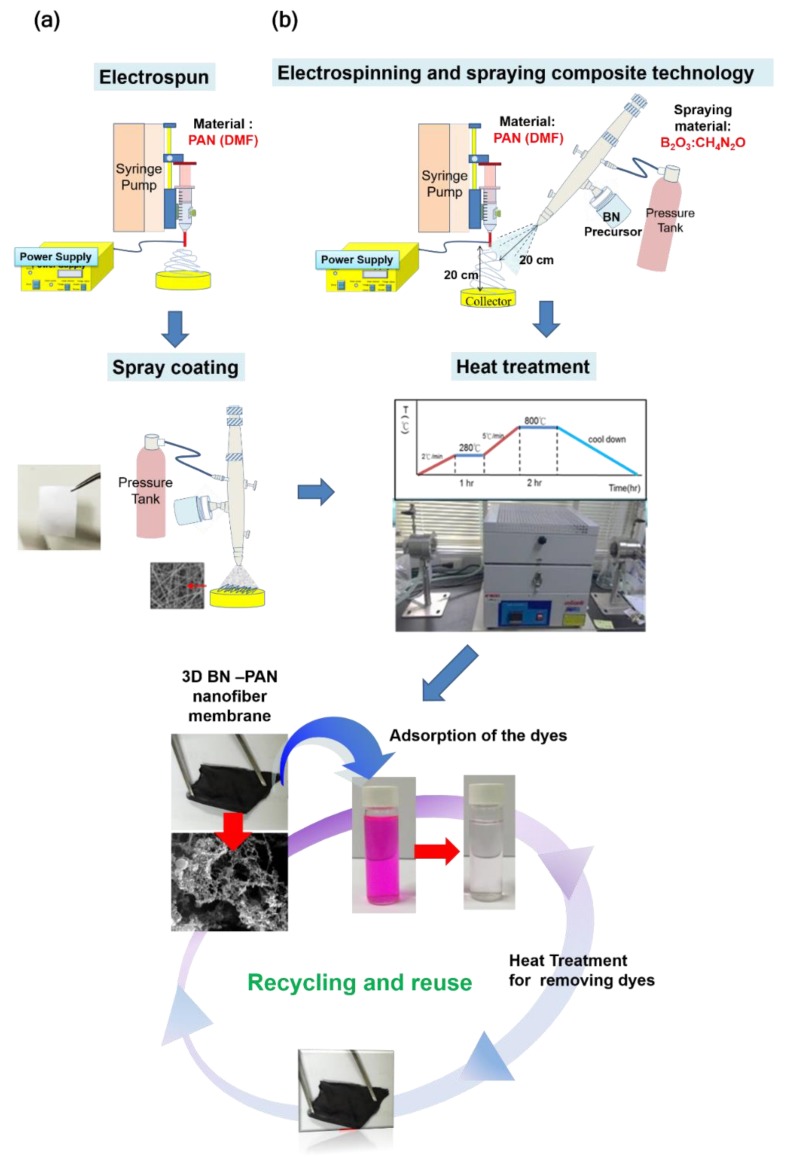
The process to produce the boron nitride (BN)- polyacrylonitrile (PAN) electrospinning (ES) polymer nanofibers (NFs): (**a**) spray-coating BN precursor to PAN NFs. (**b**) directly spray coating BN precursor and electrospinning PAN NFs at the same time.

**Figure 2 nanomaterials-09-01383-f002:**
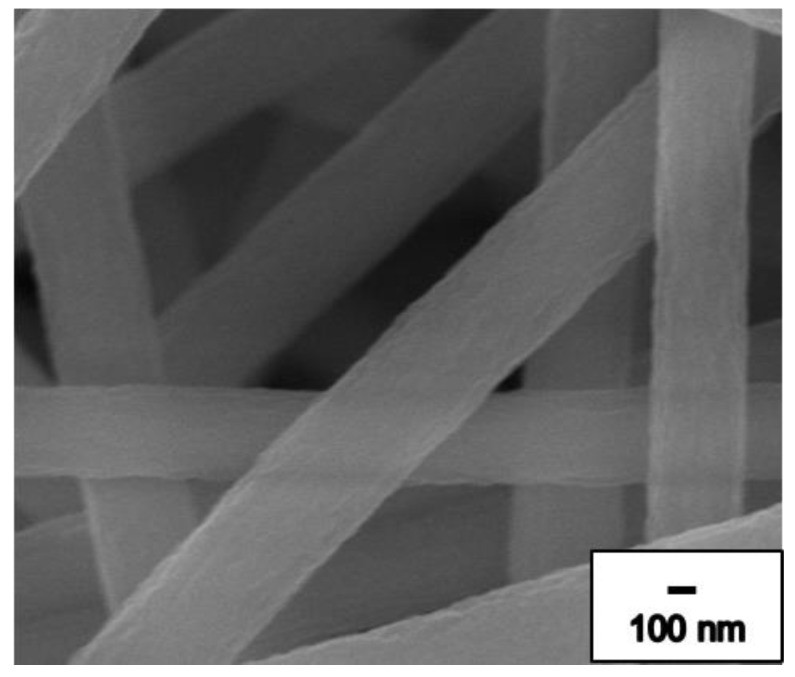
Field emission scanning electron microscope and energy dispersive spectrometer (FE-SEM) image of pure PAN ES NFs with 10,000× magnification.

**Figure 3 nanomaterials-09-01383-f003:**
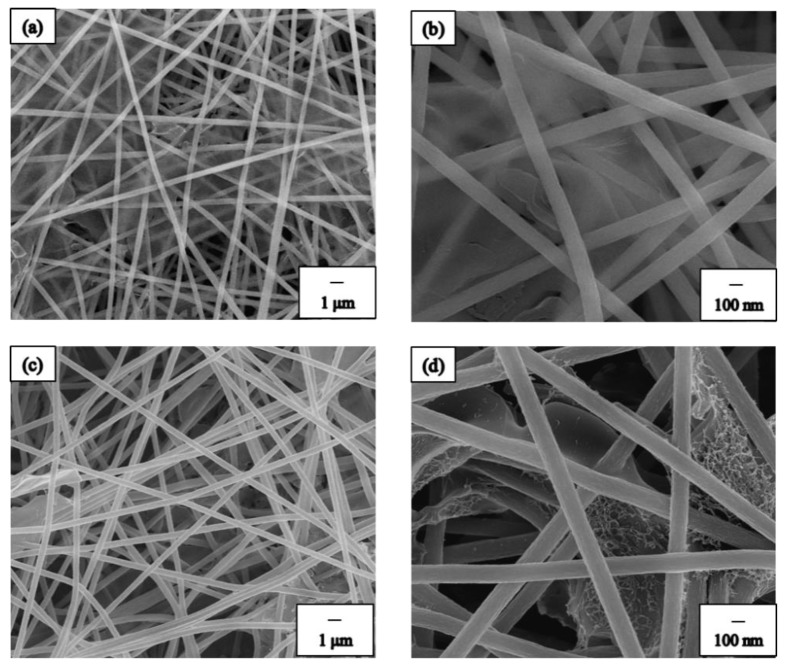
FE-SEM images of directive spray coating process: (**a**) low magnification, (**b**) high magnification; electrospinning and spraying composite process: (**c**) low magnification, (**d**) high magnification.

**Figure 4 nanomaterials-09-01383-f004:**
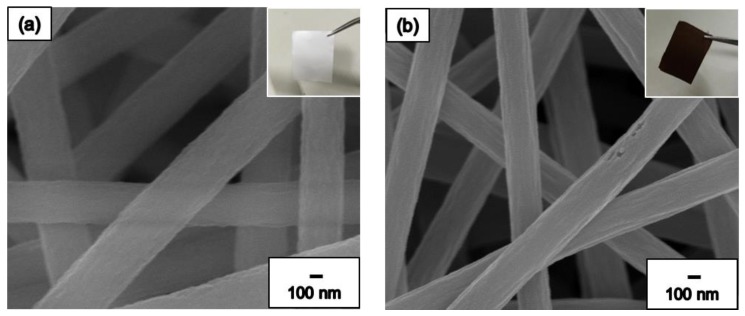
FE-SEM images of (**a**) pure PAN NF and (**b**) pure carbon NF with high magnification.

**Figure 5 nanomaterials-09-01383-f005:**
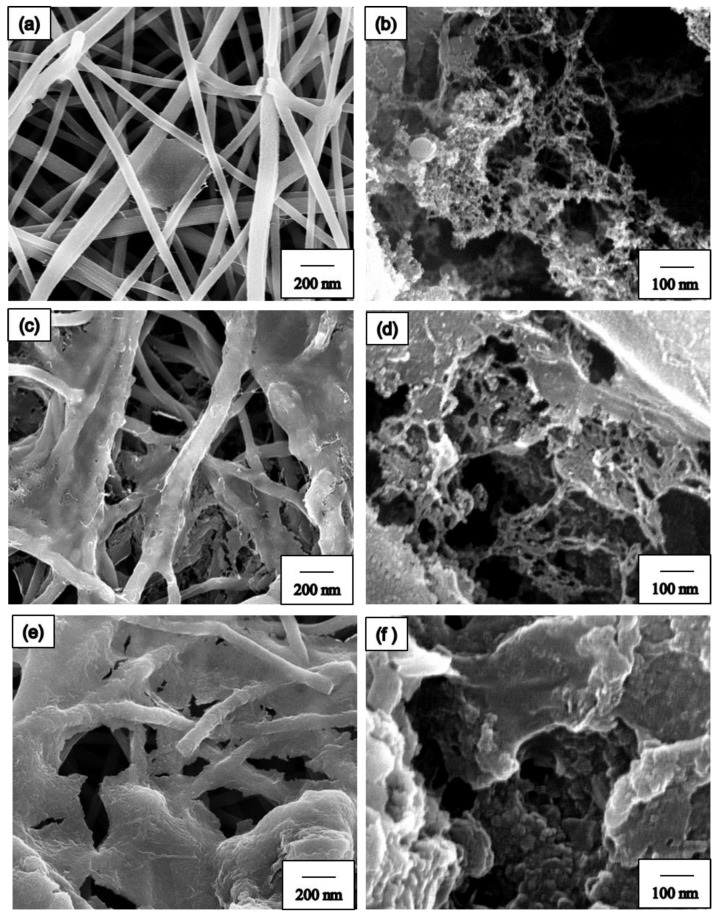
FE-SEM images of BN structure synthesized with different precursor ratio of B_2_O_3_ and CH_4_N_2_O with different molar ratio. 1:10 (**a**) low magnification, (**b**) high magnification; 1:5 (**c**) low magnification, (**d**) high magnification; 1:1 (**e**) low magnification, (**f**) high magnification.

**Figure 6 nanomaterials-09-01383-f006:**
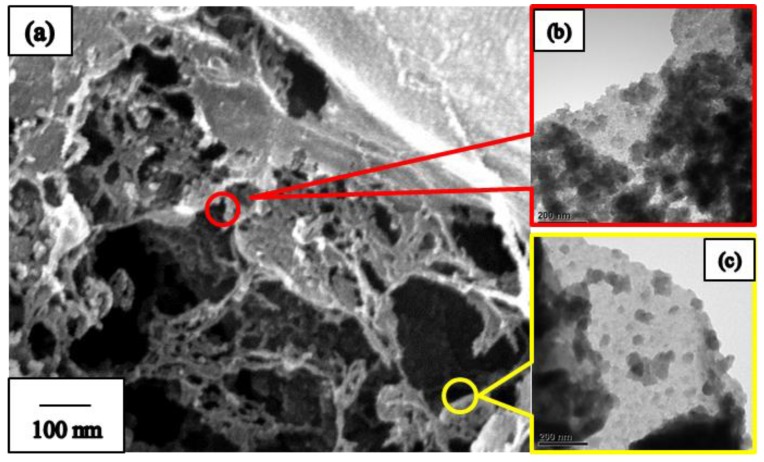
(**a**) FE-SEM image of the 3D BN porous structure, (**b**,**c**) the transmission electron microscope (TEM) images of the 2D BN nanosheets stack form the 3D porous structures on different places.

**Figure 7 nanomaterials-09-01383-f007:**
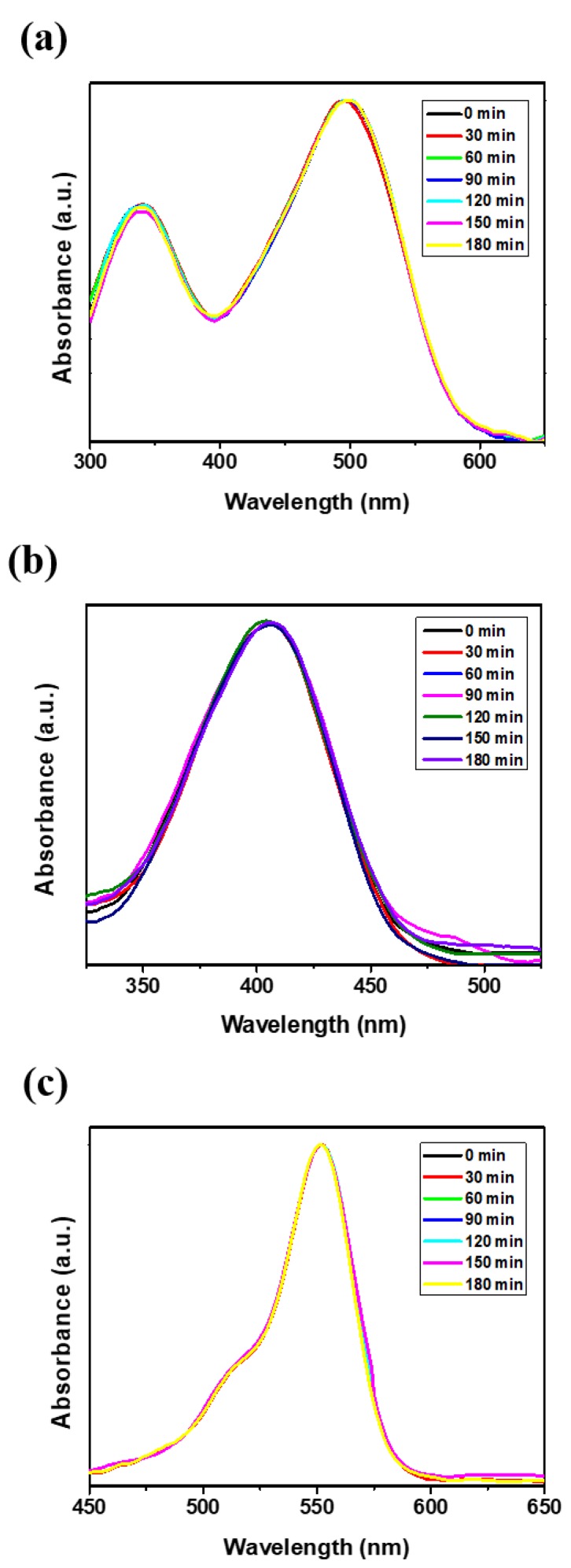
UV-vis absorption spectra of (**a**) the Congo Red (CR) aqueous solution (10 mg/L), (**b**) the basic yellow (BY) aqueous solution (50 mg/L), (**c**) the Rh B aqueous solution (10 mg/L).

**Figure 8 nanomaterials-09-01383-f008:**
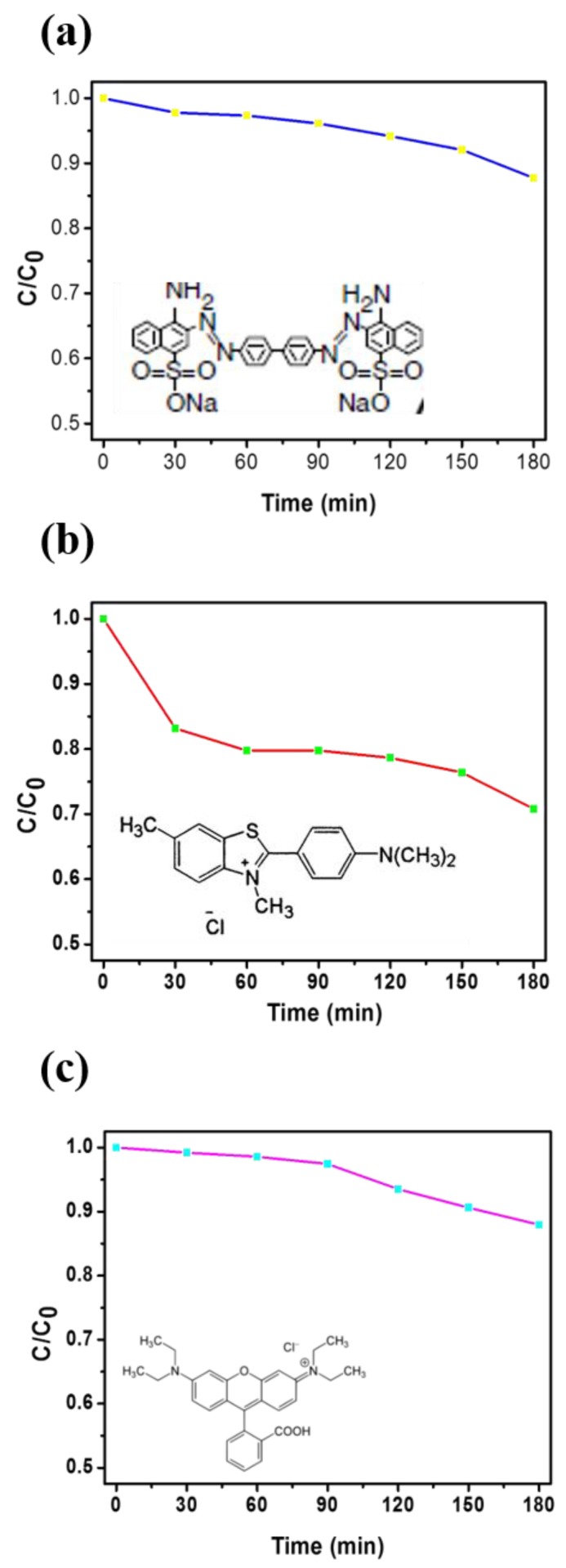
The removal efficiency of pure carbon NF membrane towards (**a**) CR, (**b**) BY, (**c**) Rh B aqueous solutions.

**Figure 9 nanomaterials-09-01383-f009:**
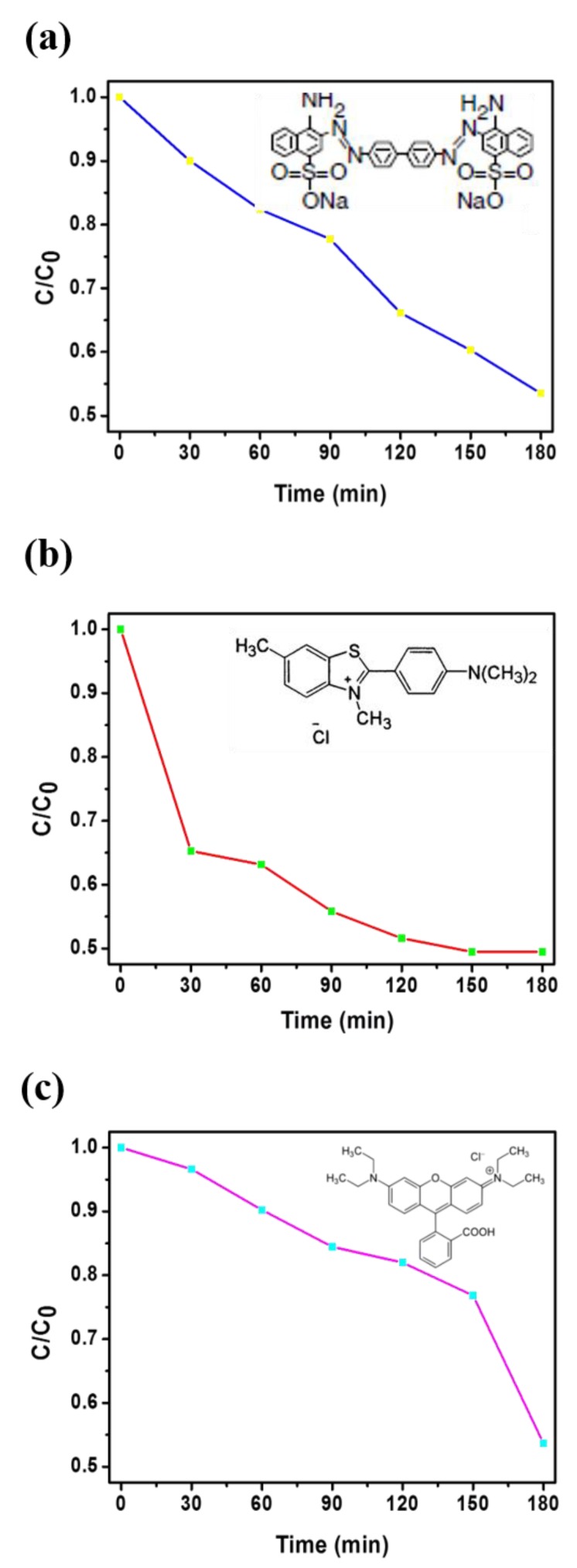
The removal efficiency of the 3D BN NF membrane in (**a**) CR, (**b**) BY, (**c**) Rh B aqueous solutions.

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
