# Peer review of "Facile 3D Boron Nitride Integrated Electrospun Nanofibrous Membranes for Purging Organic Pollutants"

_nanomaterials, 2019, doi:10.3390/nano9101383_

Round 1

Reviewer 1 Report

There are many deficiencies in the manuscript., such as:

Line 128 'mechanism  'Authors areescribing the method of preparation. Not mechanism.    

Line 141-143 --- is confusing. Explain it in more detail.

Line 160-Chemical and structuralcharacterizationof morphology.

One high resolution figure is enough to discuss the findings. Delete low resolution figures.

Line201 "We calculatedthe distributionusing low magnification SEM images.

How? Explain it.

Line214  Cyclization "to forman orderly raw-column cyclic structure"

Please defie it in few senteces and give reference.

Line206

Figure 4. Only high resolution 2 figures , 'b' and'd' are enough.

Line213-216  ' The differences between ---- denitrification processed"

It needs more explanation.  Is there any reference?

Line 23-242.  "The EDX spectrum in Fig. s(3)b-----by using our method.

What is your method?  It needs more explanation, How C, O, B an N in particular?

Line 246 -243.  The TEM images in Figs. 6(b, c) clealy------porous architecture.

Please explain it in detail.

Line 261-271.

Figure 7 is confusing (due to 7 color). All figures are identical. It would be better if author's describe it in other way.

Present manuscript can be acceted after major revision.

Reviewer 2 Report

The study investigates the effect of 3D Boron Nitride incorporated PAN ES NF membrane for dye removal by adsorption. The use of BN for adsorption applications has been widely reported but different methods of preparation produce absorbent with varying performance. In this report, an electrospinning spray coating method was used. Further comments are given below:

1) The study is not a membrane separation process but more of an adsorption process. The PAN NF film is acting like a scaffold rather than a membrane as no membrane filtration test has been performed. I would suggest the authors to avoid referring to the sample as a membrane to avoid confusion (although it could very well be used as a membrane).

2) Ref [34] is not a study of BN modified membrane for filtration. Suggest updating the references for the filtration application (Page 2 line 74). Examples of reported BN nanosheet modified membranes: 

https://doi.org/10.1016/j.memsci.2018.07.003

https://doi.org/10.1002/ange.201809126   3) Page 3 Line 115: Please check this paragraph as there are multiple typological errors in this paragraph. (double full stop, unit of pressure, and ten, thirty and sixty seconds should read 10, 30 and 60 s)   4) On several occasion BN is referred to as BN nanosheets. Successful preparation of BN nanosheet should be confirmed using Raman spectroscopy and AFM. The XPS is not sufficient to show that BN is formed (it only shows that B and N elements are available).    5) Figure 9: UV vis absorption spectra should be provided for consistency (or remove Figure 8 UV Absorption spectra)   6) It was mentioned in the abstract that the membrane can be recycled and reused for at least 10 times but results are not found. Please provide the plot of the absorption vs time for the 10 cycles. Please also provide the experimental procedure for this experiment. Was fresh feed (same initial concentration) used in each cycle?    7) Following the above and Figure 8-9, the adsorption does not lead to the complete removal of the dye. Did the author perform longer adsorption experiment until near-complete removal of the dye? Looking at Figure 9 (a) and 9 (c), complete adsorption could potentially be achieved if experiment was extended to 360 min.   8) Figure 9c:  Could the authors explain why there is a dramatically drop at time 150 min and is this result reproducible?   9) It was also observed that the BN modified sample performed much better for Rh B aqueous solutions (Comparing 8f to 9c). Could the authors provide an explanation? 

Round 2

Reviewer 1 Report

Authors improved the manuscript  and accept it for publication.

Reviewer 2 Report

Happy with the responses but please also include the discussion in the manuscript, not just in the response to the reviewer:

1) Response 6 and the figure. Please include these discussions in the manuscript.

2) Response 7. Please include the figure to show the full complete removal of dye over time (450 min). Only one sample is needed to demonstrate this.

3) Response 8 and 9. Please include these discussions in the manuscript.

These discussions are important and necessary otherwise the work is not complete. Taking response 6 as an example, the authors clearly mentioned in the abstract that the membrane can be recycled and reused for at least 10 times. Yet, this is completely missing in the main text. It is expected that there are results and discussion of the recyclability / long term durability and experimental procedures should also be provided. 
